# Assessment of Oral Health in a Child Cohort of a Rural Zone of Ethiopia

**DOI:** 10.3390/children10111824

**Published:** 2023-11-17

**Authors:** Luca Mezzofranco, Francesca Zalunardo, Andrea Magliarditi, Antonio Gracco

**Affiliations:** 1Department of Neurosciences, University of Padua, Via Giustiniani 2, 35122 Padova, Italy; francesca.zalunardo@studenti.unipd.it (F.Z.); antonio.gracco@unipd.it (A.G.); 2Private Practice, 35122 Padova, Italy; andreamaglia@yahoo.it

**Keywords:** pediatric dentistry, oral health, Ethiopia

## Abstract

Health conditions in a developing state such as Ethiopia are precarious; in addition to the extreme spread of infectious diseases such as AIDS, oral health is also affected by the scarce knowledge and possibility of treatment. This analysis considered 77 children aged 7 to 11 from a primary school in Guraghe in Ethiopia. The presence of plaque, calculus, and caries was evaluated. For each caries, the affected tooth and the site of onset were considered. Plaque was detected in 39% of the children, calculus in 22%, and dental caries in 48% of the patients. The cavities were found to be equally distributed between the two arches, with a prevalence of location in the deciduous teeth and in the occlusal area. The implementation of home oral hygiene education and the training of health professionals who successfully promote oral health is necessary.

## 1. Introduction

The 1948 WHO Constitution defined health as complete physical, mental, and social well-being, not just the absence of disease. This holistic view shifted healthcare’s focus to overall well-being and recognized the influence of various factors like biology, lifestyle, and social environment. It led to comprehensive healthcare approaches, addressing the social determinants of health and integrating mental health into primary care. Health promotion efforts emphasized education and empowerment, while research methodologies embraced interdisciplinary collaboration. Criticisms exist regarding the idealistic nature of “complete” well-being and the responsibility placed on individuals. Nevertheless, the definition sparked discussions on health equity, social inequalities, and efforts to create a healthier world [1].

Ensuring the health of the population is a fundamental objective for the government of all countries. Nowadays, many countries focus on improving health for all and reducing health inequalities through improved leadership and governance for health. They focus on major health problems and priority areas [2].

Attention to health also involves oral wellness with the aim of a condition that allows for a good chewing function, phonation, and smile [3].

According to the World Health Organization’s definition, periodontal health should be characterized by the absence of inflammatory periodontal disease, enabling individuals to function without experiencing any negative consequences, whether physical or mental, due to previous disease. However, this definition may not be fully applicable in clinical practice, as even in healthy individuals, there are inflammatory cells present at the periodontal level. Therefore, it is more clinically acceptable to consider periodontal health as the absence or minimal presence of clinical inflammation in a periodontium that exhibits normal support.

Clinical periodontal health is determined by assessing various parameters, including gingival inflammation, probing depths, clinical attachment levels, and radiographic evidence of bone loss. A periodontium with no signs of inflammation, such as gingival redness, swelling, or bleeding on probing, is considered clinically healthy. Additionally, the absence of pathological pocket depths and the presence of stable clinical attachment levels indicate periodontal health.

In conclusion, while the World Health Organization’s definition of periodontal health highlights the absence of inflammatory periodontal disease, the clinical application of this definition is limited due to the presence of physiological inflammatory cells in a healthy periodontium. Therefore, a more practical approach is to define periodontal health based on the absence or minimal presence of clinical inflammation and the preservation of normal periodontal support. This concept of clinical periodontal health provides clinicians with a framework to assess and manage periodontal conditions effectively [4].

In a randomized clinical trial, researchers conducted a study to evaluate the impact of oral health education on oral health knowledge, attitudes, and plaque scores among third-grade students. The study involved comparing a group of students who received oral health education with a control group that did not receive any intervention.

The findings of the study demonstrated significant improvements in oral health knowledge and attitudes among the students who received the oral health education program. These students exhibited a greater understanding of oral hygiene practices, dental care, and the importance of oral health. They also displayed more positive attitudes towards oral health, emphasizing the significance of regular toothbrushing, dental check-ups, and a healthy diet.

Additionally, the study assessed plaque scores as an objective measure of oral health. The results showed that the students who received the oral health education had significantly lower plaque scores compared to the control group. This suggests that the education program effectively influenced the students’ oral hygiene behaviors, resulting in reduced plaque accumulation.

The study highlights the importance of oral health education in improving oral health outcomes among young students. By providing children with the necessary knowledge and fostering positive attitudes towards oral health, such programs can empower them to adopt proper oral hygiene practices and make informed decisions about their oral health. The findings support the implementation of oral health education initiatives in school settings, with the goal of promoting better oral health in the target population [5].

The (oral) health of the population, despite being a primary worldwide target, cannot be guaranteed in an acceptable way in many countries. The management of health policies depends strictly on the conditions of the economic development of the individual countries [6].

A qualitative study performed between 2015 and 2020 in the Jimma Zone, a rural area in Ethiopia, was undertaken to explore community perceptions and experiences related to health and health inequality [7].

The definition of health for the study participants was the absence of disease or symptoms. The most frequent references to the pathologies were: AIDS, diarrhea, gastritis, and the common cold. Mental issues are also recognized as a problem. Health inequalities were viewed as community issues, primarily emanating from a lack of knowledge or social exclusion [7].

The comparison between rural and urban areas has also been proposed in several studies that evaluate oral health conditions in Nigeria. A study on oral health found out that periodontal disease with deep pocketing occurs in Nigerians at an early age, the prevalence being 15–58% in those aged above 15 years. Caries experience is moderate in some urban communities. Although the mean DMFT is below 4 in most communities, the restorative index is extremely low, most carious teeth remaining unrestored. The higher caries prevalence in second than in first permanent molars that has been reported is most likely due to a change from a traditional to a Western-type diet. Other oral health problems include malocclusion, traumatized teeth, dental fluorosis, and oral cancer. The scanty oral health services available in the country are mainly in urban areas [8].

A study stated that in the rural area of Nigeria, the occurrence of dental caries appears to be increasing in rural Nigerian schoolchildren. The oral hygiene status was poor and gingivitis was common [9].

As regards Ethiopia, the data on oral health is scarce. In the urban area of Addis Abeba, an examination on children pointed out the fact that 48% of them did not brush their teeth and 43% brushed only once daily. The majority consumed sugary food despite knowing its relationship with dental decay. A total of 74% had between 1 and 13 dental caries and 52% showed evidence of bleeding upon brushing [10].

The aim of this study is to evaluate the oral health conditions of children in rural areas of Ethiopia, considering the presence of dental caries, plaque, and calculus.

## 2. Materials and Methods

This observational study involved data collection conducted by two general dentistry doctors who conducted joint visits to children. All children were visited by the same two operators. The research was conducted in Getche, a subdivision of the Agenna municipality in the Guraghe region of Ethiopia. The class registers of the country’s primary school were utilized to ensure that all school-age children in the city could be included in the study.

The total number of children available for the study was 77, with ages ranging from 7 to 11 years old. Once the list of children was obtained, each child was individually summoned for a visit, during which a probe, mirror, and periodontal probe were used. A clinical dental record was compiled for each child, documenting the presence of plaque, tartar, or caries.

Regarding plaque, each individual tooth was circumferentially probed to determine if the probe remained coated with plaque or not, following the Silness and Loe Plaque Index (PI) criteria. The PI is a commonly used index to evaluate the amount of plaque on teeth. It ranges from 0 to 3, with higher scores indicating a greater amount of plaque. The probe was gently applied to each tooth surface, and the presence or absence of plaque was recorded.

Calculus, also known as tartar, was visually highlighted and assessed with the aid of a periodontal probe. Calculus is a mineralized deposit that forms on teeth due to the accumulation of plaque over time. It can contribute to gum disease and other oral health problems. The periodontal probe was used to detect the presence of calculus by gently scraping the tooth surface and feeling for rough or hard deposits.

Caries, commonly known as dental cavities, were identified through visual inspection and probing. Caries result from the demineralization of the tooth structure caused by acid-producing bacteria. The affected tooth and its location were carefully recorded for each instance of caries. The DMFT (decayed, missing, and filled teeth) index was used.

To ensure consistency in the data collection time, the visits were conducted over a period of 5 consecutive days. The operators followed a standardized protocol to ensure uniformity in the examination and recording of dental conditions. They were trained to use the same techniques and criteria for assessing plaque, calculus, and caries in order to minimize variability.

Once the data collection phase was completed, the collected information was compiled and analyzed. Continuous data, such as age, were summarized using the median and interquartile range (IQR), which provides a measure of the spread of the data. Categorical data, such as the presence of plaque, tartar, or caries, were presented as frequencies and percentages.

Statistical analysis was performed to examine any potential associations or differences between variables. The Mann–Whitney test, a non-parametric statistical test, was employed to compare continuous data between two groups. This test is appropriate when the data do not follow a normal distribution or when the assumptions for parametric tests are not met. Categorical data were analyzed using the chi-square test or Fisher’s exact test, depending on the sample size and expected cell frequencies. These tests allow for the evaluation of associations between categorical variables.

All statistical tests were two-sided, and a *p*-value less than 0.05 was considered statistically significant. The statistical analysis was conducted using R software, version 4.0 (R Foundation for Statistical Computing, Vienna, Austria), which is a powerful and widely used tool for statistical analysis in research.

## 3. Results

The present analysis encompassed a comprehensive assessment of the oral health status among a cohort of 77 children (36 males and 41 females) aged 7 to 11 years attending an elementary school in the Guraghe region of Ethiopia. However, it is important to note that four children from the same school could not be included in the evaluation due to various reasons.

Our findings revealed that 39% of the children exhibited plaque accumulation (Figure 1). Interestingly, we observed a significant association between the presence of plaque and age (*p* = 0.01), with a median age of 8 years (IQR 7–9) among children with plaque, compared to 9 years (IQR 8–10) among those without plaque. However, we found no significant association between plaque and sex (*p* = 0.99) (Figure 2).

Moreover, calculus was detected in 22% of the children (Figure 1). Surprisingly, we observed no significant associations between calculus and either sex (*p* = 0.18) or age (*p* = 0.32) (Figure 3).

Among the participants, 48% presented with caries, with a total of 73 cavities identified (median two cavities, IQR 1–2) (Figure 1). The distribution of the cavities revealed that 52% occurred in the upper arch and 48% in the lower arch. The majority of caries affected deciduous teeth (70% of cases), with deciduous molars being the most frequently involved teeth (60% of cases). The distribution across different tooth surfaces showed that 44% of the cavities were located in occlusal sites, 32% in mesial sites, 16% in distal sites, 4% in disto-occlusal sites, and 4% in disto-buccal sites.

Furthermore, our analysis indicated a significant association between the occurrence of cavities and a younger age (median 7 [IQR 7–9] vs. 9 [IQR 9–10], *p* = 0.002), while no significant association was found between cavities and sex (*p* = 0.31) (Figure 4).

Taken together, the prevalence rates estimated from our data suggest a considerable burden of oral health issues in the studied population, with 39% (95% CI 28 to 51%) experiencing plaque, 22% (95% CI 14 to 33%) exhibiting calculus, and 48% (95% CI 37 to 60%) presenting with caries.

## 4. Discussion

In Ethiopia, similar to many other countries in Africa, oral health receives limited attention. The availability of dental facilities is scarce, and the cost of treatments is often beyond the reach of many individuals. The lack of a comprehensive health system further exacerbates the challenges faced by the population. Consequently, a significant portion of the population, including young individuals, exhibits high DMFT (decayed, missing, filled teeth) indices. This can be attributed to the absence of adequate tools and resources for oral hygiene education at home. Without proper education and access to preventive measures, individuals are unable to maintain good oral health practices, leading to a higher prevalence of dental issues. Addressing these challenges requires comprehensive strategies that encompass improving access to dental services, implementing oral health education programs, and strengthening the healthcare system to provide basic oral health services for the population. Research published in 2020 evaluated the prevalence of caries in patients who went to the university clinic of Gondar (Northeast Ethiopia) and showed that caries was present in 23.64% of the patients, with a strong association with diabetes, poor education, irregular visits to the dentist, and poor brushing habits [11].

A recent study conducted in Debre Berhan, a city in central Ethiopia, revealed that 34.1% of children attending primary school had dental caries. The findings of the study also indicated a significant correlation between the consumption of sugary substances, specifically tea, and the occurrence of dental caries [12].

This suggests that the progressive westernization of diets among both children and adults residing in Ethiopian cities has contributed to the rise in tooth decay cases. As traditional dietary habits are gradually replaced by the consumption of processed foods and sugary beverages, the risk of developing dental caries increases. These dietary shifts, combined with the limited access to oral healthcare and oral hygiene education, pose significant challenges to oral health in urban Ethiopian communities. Efforts to address this issue should focus on promoting oral health awareness, implementing preventive measures, and encouraging healthier dietary choices to combat the growing prevalence of tooth decay in the population.

It has been shown that although there is awareness regarding bad eating habits such as the overconsumption of drinks or very sugary foods or foods rich in fat, the Ethiopian population (both urban and rural) remains indifferent to the physical implications that a bad diet can cause [13].

The variety of the Ethiopian diet appears to be scarce, partly linked to the concern of being able to obtain food [14]. The variety of food increases proportionally especially in rural areas with food security [15].

A study comparing malnutrition and the variety of diets of young Ethiopians in urban and rural areas shows that there are no strong differences between the types of diets, but that malnutrition is slightly more concentrated in rural areas [16].

In comparing the findings of our study conducted in a rural area, in which the prevalence of caries was observed in 48% of the patients, with a study conducted in the capital city of Addis Ababa, in which 74% of the patients were found to have cavities, a notable difference in the dietary patterns and consumption of sugary substances becomes apparent. This suggests that the degree of Westernization and urbanization in Addis Ababa may contribute to a higher prevalence of caries due to changes in dietary habits and increased access to processed foods and sugary drinks. These findings align with previous research highlighting the association between dietary factors, particularly the consumption of sugars, and dental caries.

Furthermore, when comparing our results with studies conducted in another urban area, albeit not the capital and without the same level of Westernization as Addis Ababa, it is evident that the percentage of caries in the rural area is higher than that in the city, which reported a prevalence of 34.1%. This discrepancy may be attributed to various factors, including differences in access to oral healthcare services, oral hygiene practices, socioeconomic status, and cultural factors influencing dietary choices.

The higher prevalence of caries in the rural area could be attributed to the limited access to dental care services, lack of oral health education, and inadequate preventive measures. It is crucial to address these disparities by implementing comprehensive oral health promotion programs, emphasizing the importance of regular dental check-ups, proper oral hygiene practices, and reducing the consumption of sugary foods and drinks.

The findings from these studies highlight the urgent need for oral health initiatives targeting both urban and rural populations in Ethiopia. Efforts should focus on raising awareness about the importance of oral hygiene, providing accessible and affordable dental services, and promoting healthy dietary habits. Collaborative efforts involving oral health professionals, policymakers, educators, and communities are essential to address the oral health challenges and reduce the burden of dental caries in Ethiopia.

The comparison of our study’s results with studies conducted in different settings reveals the impact of diet and Westernization on the prevalence of caries. The higher occurrence of caries in the capital city and the disparity between rural and urban areas underscore the importance of implementing effective oral health interventions tailored to the specific needs of different populations. By addressing the underlying factors contributing to the development of caries, such as dietary patterns and access to oral healthcare, Ethiopia can strive towards improving oral health outcomes and promoting overall well-being in its communities [10,12]. From this, it can be deduced that, in the urban area, the possibility of having an education that also includes the basic rules of oral hygiene is easier to obtain together with dental care.

A study carried out on a population of Ethiopians evaluated the variations in their periodontal and caries conditions over 6 years: a progressive worsening of periodontal conditions and an increase in cavities was highlighted, demonstrating the poor level of oral hygiene [17].

A recent study carried out on the Ethiopian population showed that 42.4% of the participants were affected by periodontal disease [18]. The accumulation of plaque and calculus with consequent gingivitis represent the prelude to periodontal disease. In our study, 39% of the children had plaque and 22% of the patients had calculus accumulations.

The results of this study clearly show the need to provide oral hygiene education in young patients who, with simple brushing, can control the accumulation of plaque and, therefore, the formation of tartar and subsequent periodontal problems and the onset of caries.

However, a recent study shows that the knowledge of the operators who deal with oral care in Ethiopia is scarce and needs a strong implementation [19]. The problems of poverty, the lack of adequate tools for oral hygiene, but also of trained professionals who can contribute to spreading good oral hygiene habits from a young age represent a big burden for the development of health in Ethiopia [20].

Our analysis appears to be the first dental study conducted in the Guraghe region. The authors hope to be able to repeat the experience in the same area in the future to verify whether the oral hygiene instructions that have been provided to the population have contributed to improving the oral health conditions of the inhabitants.

The study is limited by the small number of available patient cohorts. It would have been useful for the statistical analysis on which the study was based to have available a larger number of children.

## 5. Conclusions

This study highlights the importance of adopting oral hygiene protocols and training healthcare practitioners to oversee patient control and care. This is crucial for enhancing periodontal conditions and reducing the prevalence of caries in both rural and urban settings. It is imperative to provide patients with comprehensive education from an early age, promoting positive habits that foster long-term oral health. Furthermore, there is an urgent need to disseminate guidelines that enable individuals to maintain optimal oral hygiene practices and adhere to a suitable diet as much as possible. These measures are essential in combating oral health issues and improving the overall well-being in communities.

To achieve these goals, collaborative efforts among oral health professionals, educators, policymakers, and the public are paramount. By prioritizing oral health education and implementing effective preventive measures, we can strive to create healthier societies with reduced burdens of oral diseases. By raising awareness and promoting proper oral hygiene habits, we can empower individuals to take control of their oral health and make informed decisions about their dental care.

In conclusion, the adoption of oral hygiene protocols, comprehensive education, and the dissemination of guidelines are vital steps towards improving oral health outcomes. The engagement of all stakeholders is crucial in achieving these objectives and ensuring the long-term well-being of individuals and communities. By addressing oral health issues proactively, we can contribute to the creation of healthier populations and promote overall health and quality of life.

## Figures and Tables

**Figure 1 children-10-01824-f001:**
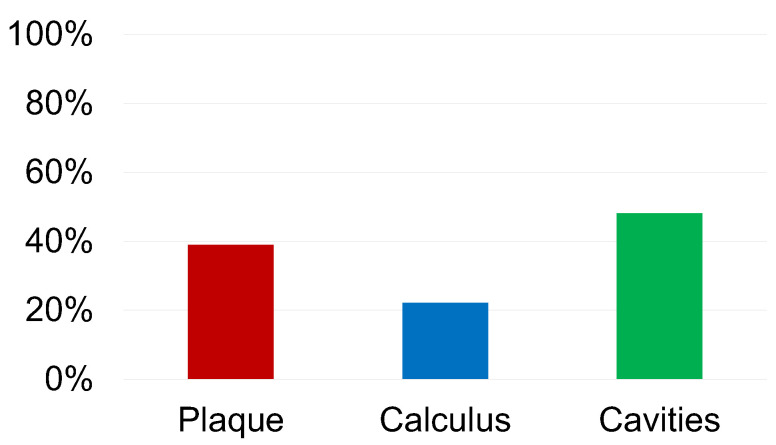
Displays prevalence percentages of plaque, calculus, and cavities in the examined sample.

**Figure 2 children-10-01824-f002:**
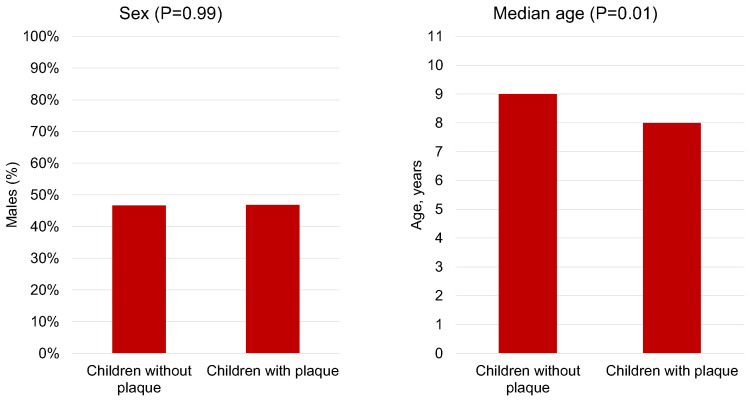
Prevalence percentages for plaque, stratified by sex and age.

**Figure 3 children-10-01824-f003:**
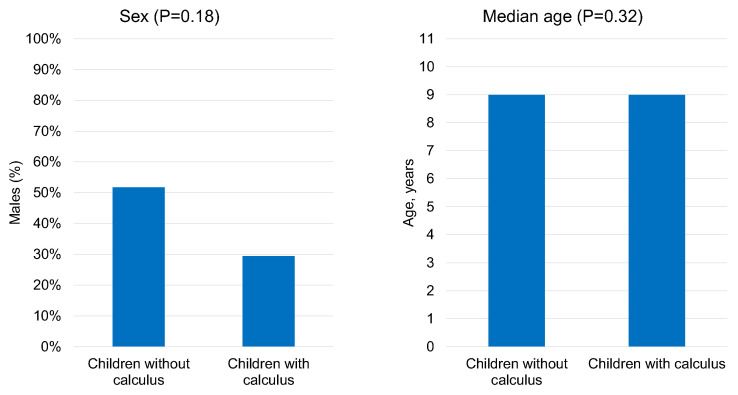
Prevalence percentages for calculus, stratified by sex and age.

**Figure 4 children-10-01824-f004:**
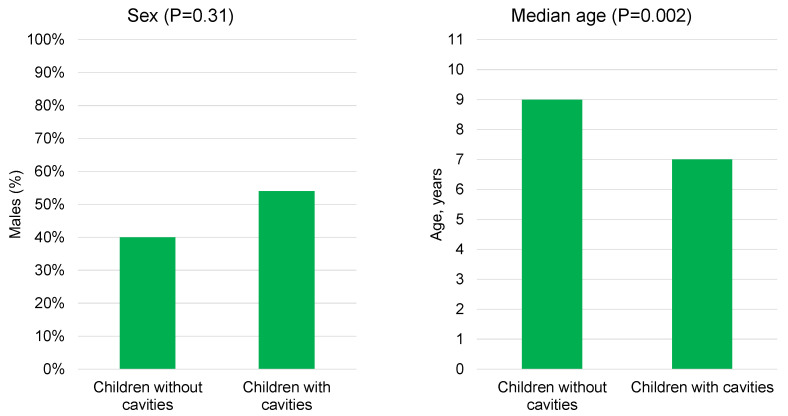
Prevalence percentages for cavities, stratified by sex and age.

## Data Availability

Data are contained within the article.

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
