# Peer review of "Assessment of Oral Health in a Child Cohort of a Rural Zone of Ethiopia"

_children, 2023, doi:10.3390/children10111824_

Round 1
Reviewer 1 Report
Comments and Suggestions for Authors
The paper "ASSESSMENT OF ORAL HEALTH IN A CHILDREN COHORT OF A RURAL ZONE OF ETHIOPIA" aims to evaluate the oral health conditions of children in rural areas of Ethiopia considering the presence of dental caries, plaque, and calculus.
The introduction is too long and confusing.
Page 5 lines 201-216 I guess are double information, like an abstract, and should be removed.
The discussion should comprise more studies related to the present results.
I suggest adding new and relevant references.
The limitations of the study should be revealed.
The conclusion should be more clinically focused.
Author Response
Good morning,
we have shortened the introduction and removed lines 201-216. Changes were also made to the discussion and conclusions part.
Reviewer 2 Report
Comments and Suggestions for Authors
Overall comments: It is simple study with an aim to describe the prevalence of periodontal conditions and dental caries among children in rural area of Ethiopia. The analysis seems appropriate for the data presented. But introduction went in detail about few ideas and studies but failed to provide any context on how do they relate to current study. No frequency counts were shared in the results. Please refer to the below comments and necessary corrections as deemed necessary.
Specific comments:
Abstract:
Line 22-24: “The progressive westernization of the diet has led to an increased consumption of sugary drinks and a consequent increase in tooth decay; the lack of oral hygiene habits and dental facilities favor the high prevalence of plaque and tooth decay in children” It may be true. However, the data shared in study only analyses the current status of the children in Getche, Ethiopia but no other data was shared to compare the change in oral health status and dietary habits.
Introduction:
Looks very lengthy. It goes over the WHO definition of periodontal health and multiple paragraphs were presented discussing the various aspects that needs to be included in the definition of periodontal health (Line 52-90). Multiple paragraphs were used to just the discuss one study and their findings. (Line 91-115). Not clear how the definition of periodontal health and the referenced study findings add any value to the aim of the current study. Would recommend to please summarize and present how does those studies/concepts relate to the current aim of the study?
Line 52: Missing citation. Please provide a citation to the WHO definition that is being referenced here.
Line 73-76: It may be true. But please provide a citation.
Line 91: Please cite the randomized clinical trial that is being discussed here. Is it reference #5?
Line 116-118: That statement seems very generic and not clear what is being said exactly.
Line 122-124: What study is being referenced here? Is it reference #7?
Methods:
Line 166 -167: Please provide the reference to “the Silness and Loe Plaque Index (PI)” being used here.
What information was collected exactly for dental caries? DMFT/DMFS?
Results:
Please present data on what type of dentition did children have? Primary vs mixed dentition vs permanent dentition?
Figure 1, 2, 3: Would recommend modifying the graphs for sex showing the presence of plaque, calculus and decay between males vs females instead of just presence and absence of conditions just for males.
Line 235-237: If majority of cavities were found in deciduous dentition and with median ages of 8 & 9, how come premolars were the most effected tooth in the mouth? Did premolars erupt completely in all the kids? Is it premolars or primary molars please confirm?
In methods, it was described that both frequencies and percentages were used to present the data for presence of clinical conditions. However, in the results and graphs only percentages were presented with no frequency counts. So, please add the frequency counts next to percentages.
What does disto-vestibular site mean? Disto-buccal?
Discussion:
Line 336: Is it 22% of children with plaque had calculus? Or 22% of the total study participants?
Line 336-337: “it is evident that periodontal health conditions cannot be maintained in adulthood if the accumulation of plaque and calculus cannot be avoided.” Seems incomplete.
Author Response
Good morning,
We have shortened the part of the introduction and removed from the abstract the requested part on the sugar diet considered unclear. DMFT was used, correcting the fact that the most affected teeth are the deciduous molars (and not the premolars).
We have corrected the wording "disto-vestibular" to "disto-buccal". The reference of 22% of affected people refers to the total patients and has been corrected in the text.
Reviewer 3 Report
Comments and Suggestions for Authors
REVIEW REPORT
Manuscript: Assessment of oral health in a children cohort of a rural zone of Ethiopia
The aim of this study was to evaluate the oral health conditions of children in rural area of Ethiopia considering the presence of dental caries, plaque and calculus.
The topic is very interesting and the manuscript could be adequate for the journal’s readers. Nevertheless, the present version needs to be improved in several conceptual and methodological aspects before to be accepted for this journal.
1. The manuscript needs to be reviewed since there are several typos in the world document. Please review the grammar structure.
2. Please follow the STROBE Guidelines for observational studies. The authors should attach a file containing the checklist in order to guarantee the accomplishment of the rules for reporting observational studies.
3. Introduction: Please summarize the aspects related to the literature review and do not establish conclusions in the introduction. Please clearly establish the scientific rationale for this study.
4. Methods: the study design needs to be clarified. In the title, the authors mention that is a cohort, but the study is descriptive in nature and does not seem a cohort study. I would like to know if there was a follow-up for the children.
5. Please improve the titles of figures in order to be clear and explicative. Please delete “presents”.
6. Methods: It is not adequate to use the expression “conclusion”. Please rewrite this section (review strobe guidelines)
7. Discussion: the authors use frequently the word “conclusion”. Please organize the discussion in the following parts
Main findings
Interpretation of results, comparison with scientific literature
strengths and limitations
Recommendations for research and practice considering the study findings.
8. Conclusions: Please summarize the conclusions, avoid using the word “conclusion” in the text.
9. References: please review and complete the information with other important references since I consider the number of references is not appropriate for this journal
10. Please improve the presentation of the text according to the requirements of the journal (paragraph structure, spaces, typos, localization of figures, etc.)
I hope these comments and suggestions could help to improve this manuscript. Regards and best wishes
Comments on the Quality of English LanguageI recommend the authors review the whole document in order to check consistency, intelligibility, and coherence. Review some typos.
Author Response
Good morning,
we have shortened the introduction and changed the wording of the references to the images. The words "conclusion" have been removed from the materials and methods and discussion parts.
Round 2
Reviewer 1 Report
Comments and Suggestions for Authors
The paper has been improved.
Reviewer 3 Report
Comments and Suggestions for Authors
No comments